# Succession of the Abandoned Rice Fields Restores the Riparian Forest

**DOI:** 10.3390/ijerph191610416

**Published:** 2022-08-21

**Authors:** Bong Soon Lim, Jaewon Seol, A Reum Kim, Ji Hong An, Chi Hong Lim, Chang Seok Lee

**Affiliations:** 1Department of Bio & Environmental Technology, Seoul Women’s University, Seoul 01797, Korea; 2Division of Forest Ecology, National Institute of Forest Science, Seoul 02455, Korea; 3Department of Bioresources Conservation, Korea National Baekdudaegan Aboretum, Bonghwa 36209, Korea; 4Division of Ecological Survey Research, National Institute of Ecology, Seocheon 33657, Korea

**Keywords:** abandoned rice fields, riparian forest, succession, South Korea

## Abstract

The vegetation changes in the abandoned rice fields with different abandonment histories were analyzed across the country of South Korea. The successional process was confirmed by changes in vegetation profiles and species composition. The vegetation profile showed the process of starting with grassland, passing through the shrub stage, and turning into a tree-dominated forest. DCA ordination based on vegetation data showed that the process began with grasslands consisting of *Persicaria* *thunbergii*, *Juncus* *effusus* var. *decipiens*, *Phalaris arundinacea*, etc., then partially went through shrubland stages consisting of *Salix* *gracilistyla*, *S.* *integra*, young *Salix* *koreensis*, etc., and ultimately changed to a *Salix koreensis* dominated forest. In order to study the relationship between the succession process of the abandoned rice paddies and riparian vegetation, information on riparian vegetation was collected in the same watershed as the abandoned rice paddies investigated. Riparian vegetation tended to be distributed in the order of grasslands consisting of *Phragmites* *japonica*, *Miscanthus* *sacchariflorus*, *P. arundinacea*, etc., shrubland dominated by *Salix* *gracilistyla*, *S.* *integra*, etc., and a *S.* *koreensis* community dominated forest by reflecting the flooding regime as far away from the waterway. The result of stand ordination based on the riparian vegetation data also reflected the trend. From this result, we confirmed that the temporal sequence of the vegetation change that occurred in the abandoned rice fields resembled the spatial distribution of the riparian vegetation. Consequently, succession of the abandoned rice fields restored the riparian forest, which has almost disappeared in Korea and other Asian countries that use rice as their staple food.

## 1. Introduction

Succession, the sequential change in species composition or structure over time following a severe disturbance, has served as an organizing concept in ecology for more than a century [1,2]. Succession is a central theme in plant community ecology, restoration, and land management [3,4,5,6,7,8]. Old-field succession, the sequence of change in plant communities on abandoned agricultural lands, has formed a large body of ecological research [9,10,11,12], and studies of old fields continue to yield new findings of interest to community ecology [8,13,14,15,16,17,18,19,20,21,22].

Old fields abandoned after dry land farming in temperate zones naturally revert to woodland with regularity, and they are treated as templates for understanding rates and trajectories of community change. The potential for natural succession after abandonment of wetland agriculture is less certain, and available literature is scarce. In a few studies, changes in species composition in abandoned agricultural fields have been inferred [23,24,25,26,27,28] and a pattern of succession was documented [29].

Rice paddy fields of temperate eastern Asia have generally increased in acreage to meet rising demand for food. In some cases, however, even in relatively populous countries such as Korea, Japan, and China, abandoned rice paddy fields are not uncommon, particularly in less productive upland areas [26,27,29,30,31]. As a result of shifting economies, rural population exodus, and the development of high-yield rice varieties, a substantial number of traditional rice fields are no longer cultivated. In Japan, for example, hopes have been raised that abandoned rice fields can play a significant role in restoring lost wetland functions and in promoting conservation of biological diversity [24,31]. In Korea, studies on vegetation succession [27,29] and the role of vegetation as a habitat of one endangered species [30] have been conducted. However, little is known about the potential nature of these places in Korea, where most of the low-lying areas that are easy to use as mountainous country have been converted into agricultural land or cities [29,32,33,34].

In mountainous Korea, the flat and gently sloping floodplains beside rivers and streams have been traditionally used for wet rice farming [35,36]. Recently, however, rapid growth of urban areas has overtaken many old rice fields, some of which lie abandoned on the fringes of suburban districts [36]. In Korea, as elsewhere, old-field succession on dry lands follows a typical progression from herbaceous to woody vegetation [37,38,39,40,41]. However, information available on the potential for succession in fallow rice paddies is very rare. In this study, we looked for evidence that abandoned rice fields return to their original environment, the river ecosystem, through a natural successional process. We also sought evidence for reference information for river restoration from the vegetation sequence of succession comparable with that observed in the spatial distribution of riparian vegetation of the natural river.

In riparian zones, flooding is the most important disturbance controlling the establishment and development of vegetation [42,43]. Flood pulses vary in their seasonal timing, frequency, duration, and magnitude. Elevation above the water level and distance from the waterfront reflect such characteristics of flooding, thereby creating temporal and spatial gradients of a hydrological regime. Plants inhabiting riparian zones either have a remarkable tolerance for flooding via physiological or morphological traits [44,45,46] or avoid the effects of flooding through timing of their life cycle [47]. Species composition and distribution patterns in riparian zones, therefore, reflect the hydrological regime and species differences in tolerance to drought or floods [48,49,50].

Colonization and development of riparian vegetation are widely dominated by disturbance conditions generated by flow and flood regimes and related morphological processes [51]. The riparian landscape is geomorphically complex and composed of a three-dimensional mosaic of habitats of fluvial origin. The riparian landscape is composed of various riparian habitats such as forest, shrub, and wetland complexes. Geomorphic work performed during flood events coupled with riparian regeneration and plant succession provides the dynamics of geomorphic processes that lead to plant communities in different developmental stages [52,53,54].

We hypothesized that abandoned paddy fields would undergo secondary succession and the process would resemble the spatial distribution of the riparian vegetation, which is dominated by the flooding disturbance regime.

The objective of this study is to identify the vegetation sequence that appears over years in the abandoned rice fields with different abandonment histories. Furthermore, this study has another purpose, which is to reveal the ecological location of the space where the rice paddies are located by comparing the information obtained here with the spatial distribution of the riparian vegetation obtained from the natural river. Finally, the aim of this study is to explore the feasibility of information on vegetation succession obtained from abandoned rice fields as the reference information for restoration of the riparian vegetation.

## 2. Materials and Methods

### 2.1. Site Description

The abandoned rice paddies studied were selected from eight areas across the national territory of South Korea (Figure 1). The survey area was first divided into western and eastern regions, and then each divided into northern, central, and southern regions, and selected evenly from each of those six regions. Pocheon and Gwangneung, Asan and Buyeo, Naju, Gangneung, Sangju, and Pohang were selected as survey sites in the northwest, west-central, southwest, northeast, east-central, and southeast regions, respectively.

The successional stages of the survey areas tended to be early stage, dominated by herbaceous plants, mid-stage, dominated by shrubby plants and young trees, and late stage, dominated by trees. Because abandonment of rice paddies began from a place away from the residential area, the successional stage tended to be determined by distance from residential area (Figure 2).

Sites for surveying reference conditions for riparian vegetation were selected in the Hantan River, Cheolwon, Gangwon Province, the Geum River, Youngdong, Chungbuk Province, the Seomjin River, Gurye, Jeonnam Province, and the Nakdong River, from Taebaek, Gangwon Province, to Andong, Gyungbuk Province (Figure 1). The Hantan River, Geum River, Seomjin River, and Nakdong River were selected for comparison with the abandoned rice paddies located in the northwest, central-west, southwest, and eastern regions, respectively.

The Hantan River is located close to the DMZ (Demilitarized Zone) between South and North Korea and thereby has relatively intact riparian vegetation, which appears in the order of grassland, shrubland, and forest as the distance from the waterway increases. The upstream reach of the Geum River, the midstream reach of the Seomjin River, and the upstream reach of the Nakdong River are located on the remote areas that have escaped excessive artificial interferences and thus retain relatively integrate riparian vegetation like the aforementioned Hantan River.

The land use pattern of the watershed of each river reach was analyzed to prove that the conservation condition of the river selected to investigate riparian vegetation is good (Table 1). The geographical location of the surveyed reach is shown in Table 2.

### 2.2. Vegetation Survey

All plant species growing in each plot were identified, following Lee [55], Park [56], and the Korean Plant Names Index [57]. The vegetation survey was carried out by recording the cover class of plant species appearing in quadrats of 2 m × 2 m, 5 m × 5 m, and 20 m × 20 m size in grassland-, shrub-, and tree-dominated stands, respectively, installed randomly in the abandoned rice paddies and the riparian zone of each river reach selected for study [9]. The vegetation survey for the abandoned rice fields was carried out in 20, 38, 34, 94, 20, 13, 43, and 54 plots in Pocheon, Gwangneung, Asan, Buyeo, Naju, Gangneung, Sangju, and Pohang, respectively. The survey for the riparian vegetation was carried out in 35, 60, 24, and 28 plots in the Hantan, Geum, Seomjin, and Nakdong Rivers, respectively. Plant cover was recorded applying the Braun-Blanquet [58] scale. Each ordinal cover scale was converted to the median value of percent cover range in each cover class. Relative coverage was determined by dividing the cover fraction of each species by the summed cover of all species in each plot and then multiplying 100 to the value. Relative coverage was regarded as the importance value of each species [59]. A matrix of importance values for all species in all plots was constructed and used as data for ordination using detrended correspondence analysis [60]. Differences in species composition depending on successional stage in the abandoned rice paddy and between the abandoned rice field and riparian landscape were compared applying the ordination method.

Vegetation stratification was prepared by depicting the profile of stand spread in a belt transect of 10 m breadth for the abandoned rice fields and rivers selected for the study.

### 2.3. Systematization of Collected Vegetation Data into Reference Information

The data collected in the abandoned rice fields of the early, mid, and late stages, which are dominated by herbs, shrubs, and trees, could be compared to the grassland, shrubland, and forest of riparian vegetation, respectively. Therefore, to use the data as the reference information for river restoration, the data were systematized by reflecting the spatial distribution of the riparian vegetation.

Reference information of vegetation was first prepared by depicting the vegetation profile, which expresses the successional stages in the abandoned rice paddies and the spatial distribution of the riparian vegetation. Furthermore, the information for species composition was also systematized by classifying into the riparian zone, which reflects the flooding disturbance regime.

### 2.4. Statistical Analyses

Ordination was carried out based on data obtained from the abandoned rice field, riparian vegetation, and by synthesizing both datasets. Detrended correspondence analysis (DCA) was applied to reveal the difference and similarity between species composition of plant communities occurring with the progress of succession in the abandoned rice field (through time) and as a function of distance from the waterfront of the riparian vegetation (through space). Ordination was performed using PC-Ord 4.0 [61].

## 3. Results

### 3.1. Successional Stages of the Abandoned Rice Fields Expressed as Stand Profile

Stand profiles of the vegetation surveyed in the abandoned rice fields in different successional stages are shown in Figure 3. The abandoned rice field of the early successional stage stays in the grassland stage, which is composed of a *Persicaria thunbergii* community, *Juncus effuusus* var. *decipiens* community, *Phalaris arundinacea* community, and so on. The abandoned rice field of the mid-successional stage is in the shrubland stage, which is dominated by *Salix gracilistyla* community, *Salix integra* community, and young *Salix koreensis* community. On the other hand, the abandoned rice field of the late successional stage maintains a forest, which is composed of an *S. koreensis* community, *Salix chaenomeloides* community, *Alnus japonica* community, and *Acer tataricum* subsp. *ginnala* community (Figure 3).

Dominant plant communities by successional stage in each survey area for abandoned rice paddies are summarized in Table 3.

### 3.2. Changes in Species Composition Depending on Successional Stages

The results of stand ordination based on vegetation data obtained from the abandoned rice fields are shown in Figure 4. In one northwestern region (Pocheon, Gyunggi-do), stands tended to be arranged in the order of *J. effuusus* var. *decipiens* community, *P. thunbergii* community, *P. arundinacea* community, and *S. koreensis* community, moving from the right to left parts on Axis I (Figure 4).

In another northwestern region (Gwangneung, Gyunggi-do), stands tended to be arranged in the order of *J. effuusus* var. *decipiens* community, *P. thunbergii* community, *S. koreensis* community, *A. japonica* community—*A. tataricum* subsp. *ginnala* community—*S*. *chaenomeloides* community, and *Fraxinus rhynchophylla* community, moving from the right to left parts on Axis I (Figure 4).

In one central western region (Asan, Chungcheongnam-do), stands tended to be arranged in the order of *J. effuusus* var. *decipiens* community, *P. thunbergii* community, *P. arundinacea* community, *S. gracilistyla* community, *S. koreensis* community—*S*. *chaenomeloides* community, and *S. integra* community, moving from the left to right parts on Axis I (Figure 4).

In another central western region (Buyeo, Chungcheongnam-do), stands tended to be arranged in the order of *Leersia japonica* community, *P. thunbergii* community—*S. gracilistyla* community*, S. integra* community, and *S. koreensis* community, moving from the right to left parts on Axis I (Figure 4).

In the southwestern region (Naju, Jeollanam-do) (Figure 4), stands tended to be arranged in the order of *J. effuusus* var. *decipiens* community, *P. thunbergii* community—*S. koreensis* community, and *S*. *chaenomeloides* community moving from the right to left parts on Axis I (Figure 4). From these results, the difference in species composition among communities could be confirmed, but the trend of succession among communities could not be confirmed.

In the northeastern region (Gangneung, Gangwon-do), stands tended to be arranged in the order of *S. gracilistyla* community, *P. thunbergii* community, *Juncus effuusus* var. *decipiens* community, and *S. koreensis* community, moving from the right to left parts on Axis I (Figure 4).

In the central eastern region (Sangju, Gyungsangbuk-do) (Figure 4), stands tended to be arranged in the order of *P. thunbergii* community, *S. gracilistyla* community–*J. effuusus* var. *decipiens* community, and *S. koreensis* community, moving from the right to left parts on Axis I (Figure 4).

In the southeastern region (Pohang, Gyungsangbuk-do), stands tended to be arranged in the order of *Typha orientalis* community, *P. thunbergii* community, *S. integra* community, and *S. koreensis* community, moving from the left to right parts on Axis I (Figure 4).

These results usually reflected the successional trend as a change from herbaceous plant-dominated vegetation through shrub-dominated vegetation to tree-dominated vegetation, except for the Naju site.

### 3.3. Spatial Distribution of the Riparian Vegetation Based on Stand Profile

Stand profiles of riparian vegetation collected from the Hantan River, Geum River, Seomjin River, and Nakdong River are shown in Figure 5. Riparian vegetation established in those four rivers appeared in the order of grassland, shrubland, and forest as the distance from the waterway increased, and thus tended to reflect the flooding disturbance regime. In the Hantan River, *Phragmites japonica* community and *P. arundinacea* community dominated the grassland zone. The shrub zone was dominated by *S. gracilistyla* community and *S. integra* community. *S. koreensis* community, *Prunus padus* community, and *Ulmus pumila* community dominated the forest zone (Figure 5).

In the Geum River, *P. japonica*, *Phragmites communis*, *Miscanthus sacchariflorus*, *P. arundinacea*, etc., dominated the grassland zone. *S. gracilistyla*, *S. integra*, *Salix subfragilis*, young *S. koreensis*, etc., dominated the shrubland zone, and *S. koreensis*, *S. chaenomeloides*, *Morus alba*, etc., dominated the forest zone (Figure 5).

In the Seomjin River, *Persicaria nodosa*, *P. japonica*, *M. sacchariflorus*, *P. arundinacea*, *Humulus japonicas*, etc., dominated the grassland zone. *S. gracilistyla*, *S. integra*, *S. subfragilis*, young *S. koreensis*, etc., dominated the shrubland zone, and *S. koreensis*, *S. chaenomeloides*, *Morus alba*, etc., dominated the forest zone (Figure 5).

In the Nakdong River, a *P. japonica* community and a mixed community of *P. japonica* and *S. gracilistyla* dominated the grassland zone. The shrubland zone was dominated by *S. gracilistyla*, *Carex dimorpholepis*, *A. tataricum* subsp. *ginnala*, etc., and *Salix koreensis*, *A. tataricum* subsp. *ginnala*, *Juglans mandshurica, F. rhynchophylla*, etc., dominated the forest zone (Figure 5).

### 3.4. Species Composition of the Riparian Vegetation

The results of stand ordination based on the riparian vegetation data obtained from the Hantan River, Geum River, Seomjin River, and Nakdong River are shown in Figure 6. In the Hantan River, stands tended to be arranged in the order of a *P. arundinacea* community, *P. japonica* community, *S*. *gracilistyla* community, *S*. *integra* community, *Acer tattarium* subsp. *ginnala* community, and *S*. *koreensis* community from the left to right parts on the Axis I, and thus reflected the typical spatial distribution of the riparian vegetation (Figure 6).

In the Geum River, stands tended to be arranged in the order of a *P. japonica* community, *Echinochloa crus-galli* community, *M. sacchariflorus* community, and *S*. *gracilistyla* community—*S*. *koreensis* community*—S. chaenomeloides* community, from the right to left parts on Axis I, and *S*. *gracilistyla* community, *S*. *koreensis* community, and *S. chaenomeloides* community were arranged in the mentioned order from the lower to upper parts on Axis II (Figure 6).

In the Seomjin River, stands tended to be arranged in the order of a *P. japonica* community, *S*. *gracilistyla* community, *S. chaenomeloides* community, and *S*. *koreensis* community from the left to right parts on Axis I (Figure 6).

In the Nakdong River, stands tended to be arranged in the order of a *P. japonica* community, *S*. *gracilistyla* community*–S*. *integra* community, and *A. tataricum* subsp. *ginnala* community–*S*. *koreensis* community from the left to right parts on Axis I (Figure 6).

### 3.5. Comparison of Species Composition between Vegetation in the Abandoned Rice Fields and the Riparian Vegetation

As a result of ordination based on vegetation data obtained from both the abandoned rice paddies and riparian zones, the arrangement of stands of the abandoned rice fields reflected successional change over years after abandonment, and that of the riparian vegetation stands reflected the spatial distribution pattern (Figure 7). In the results, stands of plant communities established in the early successional stage of the abandoned rice fields and grassland zone of the rivers, such as *P. japonica*, *P. thunbergia*, and *J. effuses* var. *decipiens* communities, tended to be arranged far away from each other (Figure 7). However, stands established away from the waterfront in the riparian zone and stands established in the abandoned rice paddy of the late successional stage with long abandonment history were located close to each other (Figure 7). That is, there was a difference in species composition between the vegetation of the early successional stage in the abandoned rice paddy and the herbaceous plant-dominated vegetation established near the waterway in the riparian zone. However, the vegetation of both sites tended to be similar to each other as the succession progressed in the abandoned rice paddy and the distance from the waterfront increased in the riparian zone (Figure 7).

### 3.6. Feasibility of Vegetation Data Obtained from the Abandoned Rice Fields as Reference Information for Restoration of the Riparian Vegetation

The result of ordination based on vegetation data obtained from the abandoned rice fields and the riparian vegetation implies that species composition of vegetation established in the abandoned rice paddy of the early successional stage was different from that in the grassland zone of the riparian vegetation (Figure 7). However, species composition of vegetation established in the old abandoned rice fields resembled that of riparian vegetation established in the zones far from the waterfront (Figure 7). These results show that vegetation of the early successional stage in the abandoned rice fields differs from that of the grassland zone of the riparian vegetation, but the vegetation that appears as the succession progresses resembles that of the tree-dominated zone of riparian vegetation.

Considering these facts, we recommend *Salix* spp. including *S. gracilistyla*, *S. integra*, etc., which form the mid-successional stage of the abandoned rice paddy, and *S. koreensis*, *A. tataricum* subsp. *ginnala*, *A. japonica*, and *Morus bombycis*, which form the late successional stage of the abandoned rice paddy, as the candidate plant species to create the shrubland and forest zones, respectively, of the riparian vegetation (Figure 8).

The reference information systematized by synthesizing the vegetation data collected from the riparian vegetation of the abovementioned four rivers was expressed as a stand profile (Figure 8). *P. japonica* and *M. sacchariflorus*, *Salix* spp. including *S. gracilistyla*, *Salix integra*, etc., and *S. koreensis*, *A. tataricum* subsp. *ginnala*, *F. rhynchophylla*, *Alnus japonica*, etc., dominated the grassland-, shrubland-, and tree-dominated forest zones, respectively.

Comparing both vegetation profiles, those from the grassland portions of the abandoned rice paddies and the riparian zone differed, but shrub-dominated and tree-dominated zones were very similar (Figure 8).

## 4. Discussion

### 4.1. The Significance and Value of a Study on Succession

A study on succession was begun from characterizing successional patterns describing the chronosequence of vegetation along sand dunes on the shores of Lake Michigan, moving from bare sand beach, to grasslands, to mature forests [62]. A description of changes in species composition over time in old fields forms a backbone of successional studies. Abandoned fields offer several advantages for ecological research because they are worldwide, and thus allow comparison of the results from various geographical regions [10]. Therefore, Rejmanek [10] viewed old fields as a sort of ‘Drosophila’ for a study on terrestrial ecology, particularly succession.

Succession is as central to ecology as evolution in biology [63]. The study of succession covers the concepts and tools of ecosystem, community, population, and species ecology, soil science, geology, meteorology, conservation biology, and other disciplines. It has attracted human attention over the centuries and has gradually become a model for habitat restoration [29,64]. Succession has played an important role in the formation of theory and concept of ecology in the past because it represents the recovery and development of ecosystems after disruption, and today it provides significant information required for the restoration of disturbed land [8,64,65].

The study of succession, how biological communities are reorganized after natural or artificial disturbances, has become the basis of ecology and theoretical frameworks that underpin many fields of study [1,22,66,67]. Although succession is sometimes recognized as a classical theme, recent research and reviews have shown that succession continues to play a central role in modern ecological theory and applications. For example, our understanding of succession is included in theories on the combination of community and coexistence of species [68,69,70], as well as development, restoration ecology, and global change ecology [71,72]. In this regard, succession can certainly function as the basis of modern ecology.

Succession refers to a directional, predictable change in community structure over time (Figure 3 and Figure 4, [4,73]). This change is due to shifts in the presence and relative abundance of different species as time passes over years to centuries. In plant communities, succession begins when an area is made partially or completely devoid of vegetation due to a disturbance [74]. Vegetation types in a given site have been and are shaped by a suite of disturbance types that vary in both their spatial extent and frequency of recurrence [75]. The integration of disturbance events over time and space is known as disturbance regimes, and they can be characterized in a variety of ways. That is, disturbance regimes vary in extent, spatial distribution, frequency, and recurrence interval depending on the environmental conditions [76,77,78,79].

Rice fields are a kind of disclimax that has been maintained for a long time due to continuous human interference. When cultivation is stopped in these rice paddies, a succession proceeds in which the structure and species composition of vegetation change over time after abandonment [29]. Disclimax refers to the plant communities in progress of succession formed under the influence of the frequency and intensity of disturbance, that is, the disturbance regime [79,80]. Disclimax is a relatively stable ecological community often including kinds of organisms foreign to the region and displacing the climax because of disturbance, especially by humans.

Riparian ecosystems are dynamic systems found in flood-prone areas along rivers. They represent the transition between the aquatic and terrestrial ecosystems [81] and play a decisive role in riverine integrity [82]. Riparian ecosystems rely greatly on the characteristics of the flow regime (e.g., [83]) and are notably susceptible to flow regime changes (e.g., [84]). The natural inter- and intra-annual variability of the flow regime determines the highly variable fluvial disturbances to which riparian vegetation respond structurally in the medium to long term [85]. Therefore, fluvial disturbances, i.e., the disruption imposed by the seasonal sequence of river flooding and drying (particularly their intensity and spatial extent), are the main drivers of the ecological succession of riparian vegetation [86]. That is, fluvial disturbances control the creation, development, and recycling of vegetation patches [87]. Fluvial disturbances, especially floods, are the main drivers of the successional patterns of riparian vegetation. Those disturbances control the riparian landscape dynamics through the direct interaction between flow and vegetation [88]. The spatial distribution of riparian vegetation types in different developmental stages reflects the disturbance regime (Figure 5 and Figure 6).

### 4.2. Relationship between Vegetation Sere in Abandoned Rice Fields and Spatial Distribution of Riparian Vegetation

In Korea, abandonment of rice paddies was a rare phenomenon in the past, but we were able easily find a number of rice paddies abandoned at different times, as the abandonment has become more common.

A sere of plant communities on abandoned rice fields was started by grassland dominated by herbaceous plants, which are usually established around the pool based on the waterway and in the marginal subzone based on the riparian ecosystem [89]. The older rice paddies dominated by shrubby willows such as *S. gracilistyla*, *S. integra*, or young *S. koreensis* were structurally analogous to the lower subzone (shrubland) in the riparian ecosystem (Figure 3 and Figure 5). The old abandoned rice fields with well-developed tree layers resemble the riparian forest, which is established in the upper subzone [90,91] of the riparian ecosystem. Vegetation of the old abandoned rice fields is composed of *S. koreensis*, *S. chaenomeloides*, *A. japonica*, *A. tataruim* subsp. *ginnala*, etc. (Figure 3 and Figure 5). Riparian vegetation of the tree-dominated upper zone is composed of *S. koreensis*, *S. chaenomeloides*, *A. japonica*, *A. tataruim* subsp. *ginnala*, *F. rhynchophylla*, *U. pumila*, etc. (Figure 5 and Figure 6), and thus shows similar species composition to those of the old abandoned rice paddies (Figure 7 and Figure 8).

### 4.3. Spatial Range of River as a Landscape

All river systems contain three distinct, lateral riparian zones, namely marginal, lower, and upper zones. The zones are identified by disturbance regime, that is, a combination of the periodicity of the hydrological influence, marked changes in lateral elevation or moisture gradients, changes in geomorphic structure, and changes in plant species distribution or community composition along lateral gradients [90,91,92].

The marginal zone is the lowest zone and is always present in river systems, while the other two zones may not always be present. The zone is situated from the water level at low flow, if present, up to the features that are hydrologically activated for most of the year. The marginal zone along this section of the river is relatively small and rarely extends for more than one meter from the water front. The marginal zone in this section does not contain a large amount of vegetation. This is primarily due to the sandy sediment and the constant disturbance and deposition of sediment in the marginal zone. The marginal zone is also often eroded away by the river. Therefore, the riparian community is currently in a poor state [90,91].

The lower zone may be accompanied by a change in species distribution patterns. The lower zone consists of geomorphic features that are activated on a seasonal basis. The lower zone along this section of the river contains a steep slope but is subjected to annual flooding. The riparian vegetation in this zone consists primarily of grasses and herbs. The lower zone is also encroached by shrubby willows such as *S. gracilistyla* and *S. integra* (Figure 5 and Figure 8). These shrubs are adapted to the high moisture regime and annual floods [90,91].

The upper zone is characterized by ephemeral features as well as the presence of both riparian and terrestrial species. The zone extends from the lower zone to the riparian corridor. The upper zone contains geomorphic features that are hydrologically activated on an ephemeral basis. The upper zone along this section of the river contains a steep slope that contains a marked decrease in lateral slope. The upper zone is usually dominated by trees such as *S*. *koreensis*, *S*. *chaenomeloides*, *A. tataricum* subsp. *ginnala*, *A. japonica*, and so on. This zone is considered severely degraded gradients [90,91].

The riverine landscape is comprised of the stream and riparian ecosystems. When water flows in stream ecosystems over a bank, riparian ecosystems are formed [90]. A riparian ecosystem is an ecotone between aquatic and terrestrial ecosystems that consists of several fluvial surfaces, including channel islands and bars, channel banks, floodplains, and lower terraces. Goodwin et al. [90] divided the riparian ecosystem into two zones, but it becomes three zones, including bare ground (marginal zone). Zone 1, which corresponds to the lower zone, is frequently inundated, affected by the current fluvial geomorphic processes, and is at elevations that allow shallow-rooted plants to extract water from the water table. Zone 2, which corresponds to the upper zone, is formed by past fluvial geomorphic processes and is higher in elevation than surfaces in Zone 1, and deep-rooted plants can extract water from the underlying alluvial aquifer dominate vegetation. However, Zones 1 (lower zone) and 2 (upper zone) are not always present in all sections of a river corridor and the separation between them is not always distinct, often existing along a continuum rather than as a sharp boundary. Riparian vegetation, which appears in the order of grassland, shrubland, and forest depending on the distance from the waterway, is determined by different disturbance regimes depending on the topographic condition of the riparian zone [90,91]. In this respect, the sequence of vegetation corresponds to a toposequence and at the same time a chronosequence.

That the temporal sequence of vegetation change identified in abandoned rice paddies after abandonment resembled the spatial distribution of the riparian vegetation could be evidence that such rice paddies were made by transforming riparian zones. In fact, in the DMZ, where human interference was limited for more than 70 years after the Korean War, the former rice fields were covered with riparian vegetation [92].

### 4.4. Land Use Pattern and Actual Status of Rivers in Korea

Korea has a high population density (https://www.worlddata.info/population-density.php, accessed on 15 July 2022). Rice, a wetland plant, is used as a staple food, but 65% of the country’s land is mountainous, making it difficult to secure farmland to cultivate it. Rice farming in Korea has more than 5000 years of history [93].

In countries where rice is used as a staple food such as Korea, a significant portion of the riparian zone, which is a part of the riverine landscape, has been used as rice paddies. In addition, these riparian zones have been developed into cities, and we can see this fact from the soil of the urban area, the alluvial soil [32,92,94,95].

Since its use for rice paddies, the rivers have been narrowed to secure as much land as possible, and embankments have been built high to prevent flood damage. Therefore, most of the rivers in Korea have very narrow riparian ecosystems, and it is hard to find a river that has a complete spatial range [92,96].

In the Demilitarized Zone and Civilian Control Zone, which were left in the natural process for 70 years after the Korean War, abandoned rice paddies restored to riparian vegetation. According to the changes in the landscape structure of the Suip stream basin in Yanggu, Gangwon-do, which flows through the Demilitarized Zone and the Civilian Control Zone, over the past 70 years, some rice paddies in the area have remained intact, but the ones in the Demilitarized Zone and the Civilian Zone have turned into riparian vegetation [92].

By ecologically evaluating the actual conditions of the rivers in Korea, the flow diversity is low due to the drop in sinuosity due to the effects of channelization. As a result, diversity of micro-topography in waterways is reduced, and thus ecological diversity is also low. The cross-sections of most rives maintain a double terrace structure, and thereby the connectivity between the stream ecosystem and the riparian ecosystem is not natural. This double terraced structure also reduces the surface area of river channels, adding to the risk of flooding along with reducing the width of rivers due to excessive land use. Furthermore, this greatly changed the ecological characteristics of vegetation established there, influencing the water table and the disturbance regime. The high proportion of obligate upland and exotic plants in riparian vegetation zones reflects this fact. In addition to the reduction in the width and cross-sectional area of the river, these vegetation changes become aggravation factors in the reduction in the surface area of the river channel and flooding risk [34,97,98,99].

### 4.5. Necessity for Riparian Restoration

Succession is a process in which the disturbed ecosystem develops toward a state before it is disturbed through natural processes [1,2]. In the abandoned rice paddies, the fact that the vegetation of the late successional stage resembles the species composition of plant communities established in the zone far from the waterway in the spatial distribution of riparian vegetation proves that location of these paddy fields was originally riparian zones.

A map showing land use patterns in Korea represents that these rice paddies are extensively distributed along the river. From these results, it could be confirmed that the width of Korean rivers was greatly reduced. Nevertheless, as recreational use increases in the remaining river spaces, the cross-section of the river changes to a double terraced structure, and the river is experiencing a reduction in the volume of the riverine structure again [97].

In the river whose width has been greatly reduced in this way, it is expected that a risk of flooding would increase. Moreover, rivers in Korea belonging to the monsoon climate are exposed to the risk of flooding every year, and the problem is even more serious as extreme weather events become more frequent and the intensity increases due to climate change [100,101].

In this reality, ecological restoration is urgently required in the direction of securing the original spatial range of rivers whose width has been greatly reduced [102]. In fact, river restoration projects in advanced countries reflect this trend [103,104,105,106,107]. The room for the river project represents such river restoration [108]. When promoting such ecological restoration, the ecological information secured through this study could be used to set the spatial range to be secured, and furthermore, the secured vegetation information could be used as reference vegetation information required to practice the restoration.

## 5. Conclusions

Today, riparian areas are treated as one of the very important ecosystems, but most riparian landscapes in Korea have been transformed into rice paddies and urban areas. Consequently, the riparian ecosystems were destroyed or degraded and thereby the river’s extent was reduced and function was simplified. In this respect, it is urgently required to restore the degraded or destroyed riparian ecosystems due to excessive land use. However, rapid environmental changes including climate change require systematic restoration to ensure stability as well as to expand the spatial extent of rivers.

In Korea, following the global trend, river restoration projects are being carried out, but the ecological quality of the streams has not improved as the principle of ecological restoration was not adhered to. In particular, the reference information, which should be a model in ecological restoration plans and a guideline for evaluation after restoration projects, is not utilized. Information on the ecological location of abandoned rice fields obtained from this study could contribute to expanding the spatial extent of river restoration in the future. Furthermore, information on vegetation obtained in the process is expected to serve as reference information in the future, which will contribute to improving the quality of river restoration.

## Figures and Tables

**Figure 1 ijerph-19-10416-f001:**
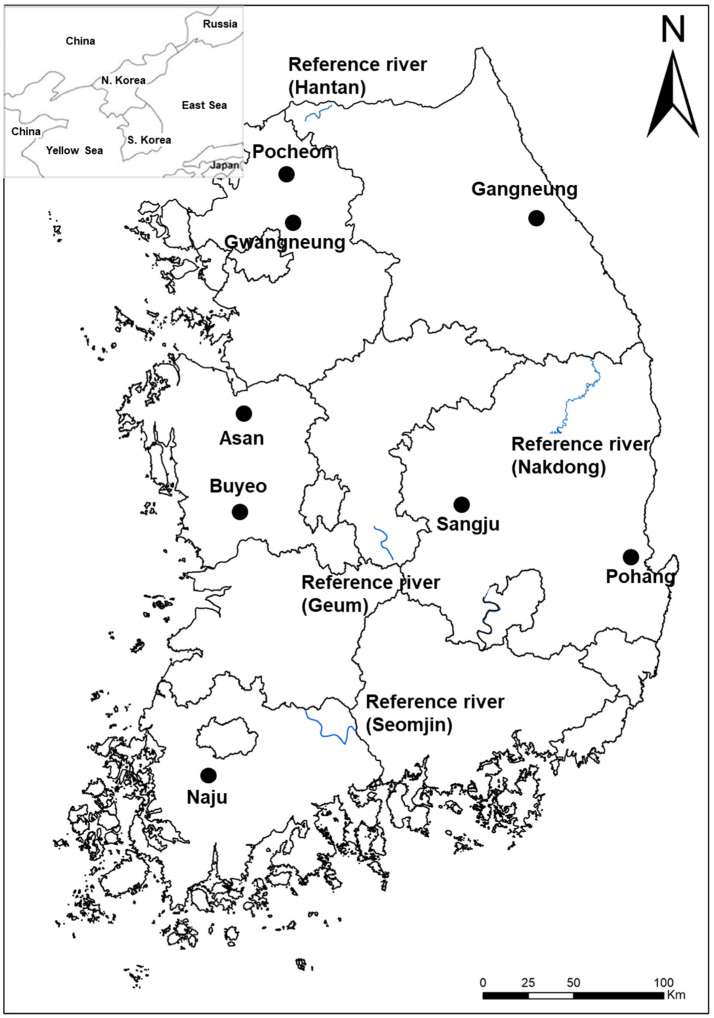
A map showing the study areas. Information for the abandoned rice fields were collected from Pocheon, Gwangneung, Asan, Buyeo, Naju, Gangneung, Sangju, and Pohang. Information for the riparian vegetation was obtained from the midstream reach of the Hantan River, the upstream reach of the Geum River, the midstream reach of the Seomjin River, and the upstream reach of the Nakdong River where riparian vegetation is conserved well.

**Figure 2 ijerph-19-10416-f002:**
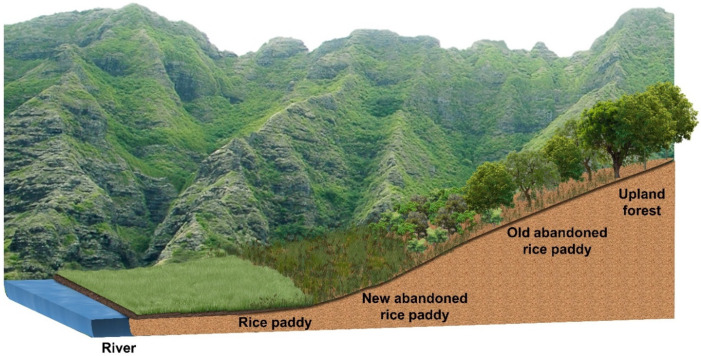
A schematic diagram showing the spatial distribution of the abandoned rice paddies and their different successional stages. Abandonment of rice paddy began in an inaccessible place away from the residential area, and the successional stage tended to be determined by the distance from residential area.

**Figure 3 ijerph-19-10416-f003:**
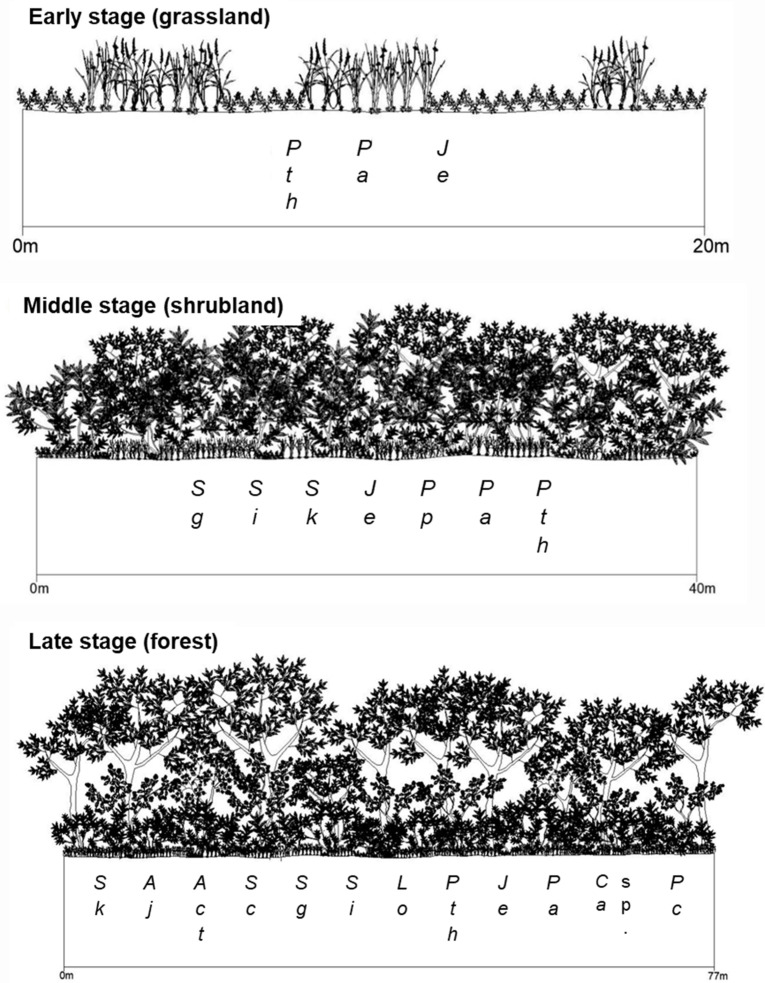
Stand profiles of vegetation investigated in the abandoned rice fields with different history*. Act*: *Acer tataricum* subsp. *ginnala*; *Aj*: *Alnus japonica*; *Ca* sp.: *Carex* sp.; *Je*: *Juncus effusus* var. *decipiens*; *Lo*: *Ligustrum obtusifolium*; *Pa*: *Phalaris arundinacea*; *Pc*: *Phragmites communis*; *Pp*: *Persicaria perfoliata*; *Pth*: *Persicaria thunbergii*; *Sc*: *Salix chaenomeloides*; *Sg*: *Salix gracilistyla*; *Si*: *Salix integra*; *Sk*: *Salix koreensis*.

**Figure 4 ijerph-19-10416-f004:**
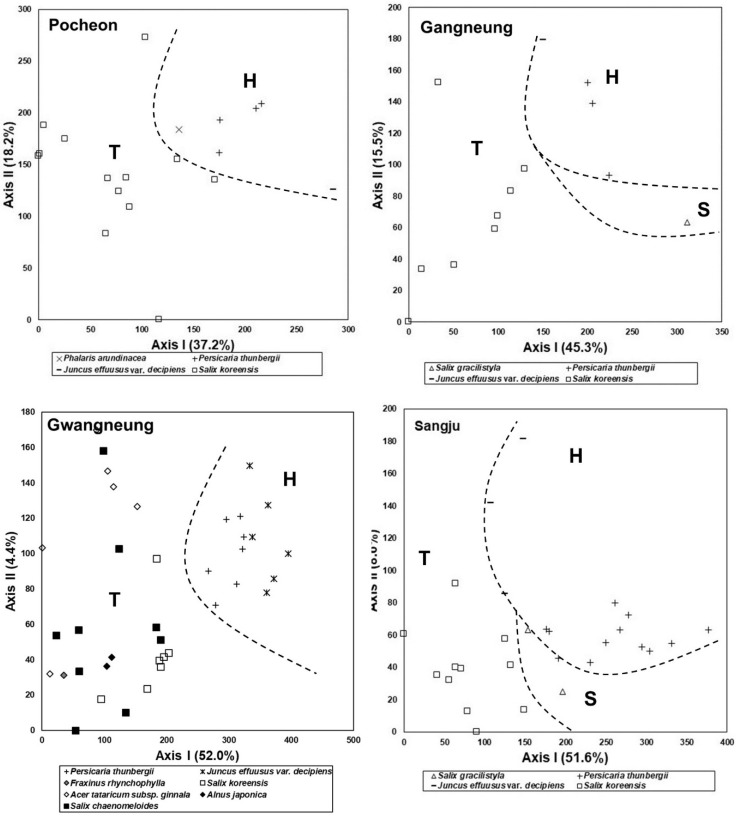
DCA ordination of stands based on vegetation data collected from the abandoned rice fields of eight areas selected across the whole national territory of South Korea. Stands tended to be arranged in the order of herbaceous plant (H)-, shrub (S)-, and tree (T)-dominated stands on Axis I or II. Arrangement of stands reflected successional change over years after abandonment of the rice paddy.

**Figure 5 ijerph-19-10416-f005:**
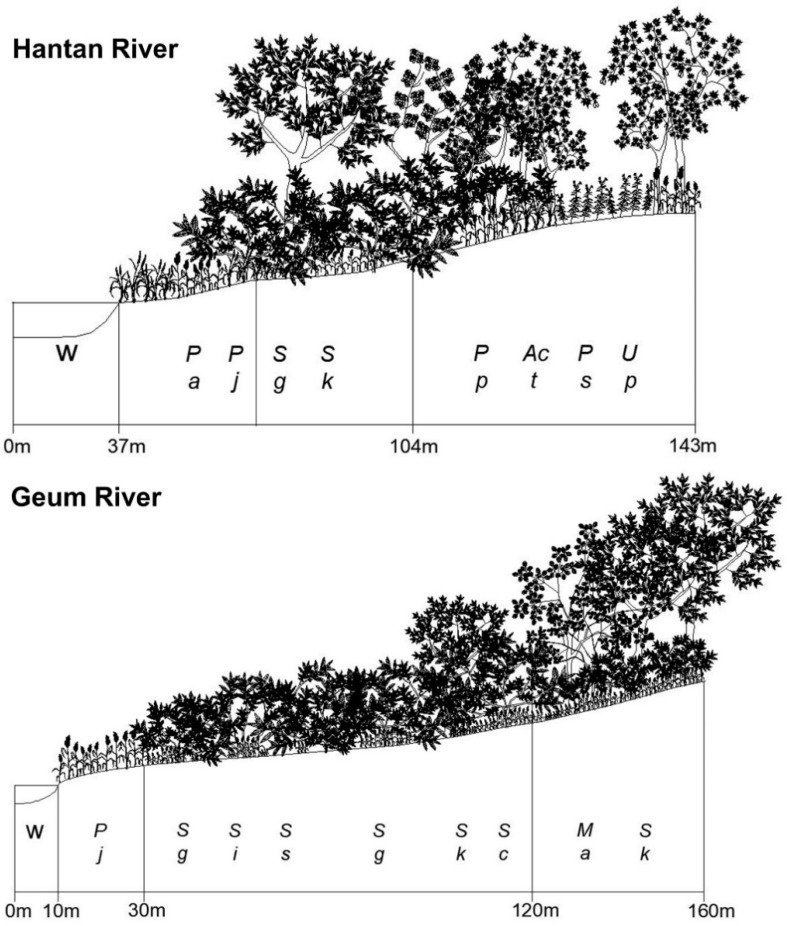
Stand profiles of riparian vegetation collected from the Hantan River, Geum River, Seomjin River, and Nakdong River. *Act*: *Acer tataricum* subsp. *ginnala*; *Aj*: *Alnus japonica*; *Cd*: *Carex dimorpholepis*; *Fr*: *Fraxinus rhynchophylla*; *Ma*: *Morus alba*; *Ms*: *Miscanthus sacchariflorus*; *Pa*: *Phalaris arundinacea*; *Pj*: *Phragmites japonica*; *Pn*: *Persicaria nodosa*; *Pp*: *Prunus padus*; *Ps*: *Polygonatum stenophyllum*; *Sc*: *Salix chaenomeloides*; *Sg*: *Salix gracilistyla*; *Si*: *Salix integra*; *Sk*: *Salix koreensis*; *Ss*: *Salix subfragilis*; *Up*: *Ulmus pumila*; W: waterway.

**Figure 6 ijerph-19-10416-f006:**
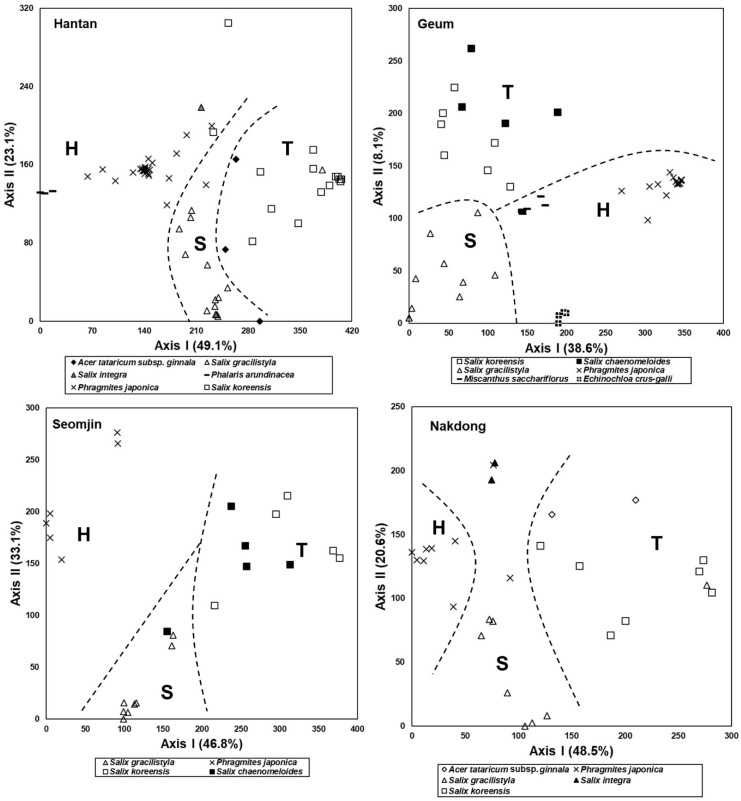
DCA ordination of stands based on vegetation data collected from the riparian zones of four rivers in South Korea: Hantan River, Geum River, Seomjin River, and Nakdong River. H: herbaceous plant dominated-stands, S: shrub-dominated stands, T: tree-dominated stands.

**Figure 7 ijerph-19-10416-f007:**
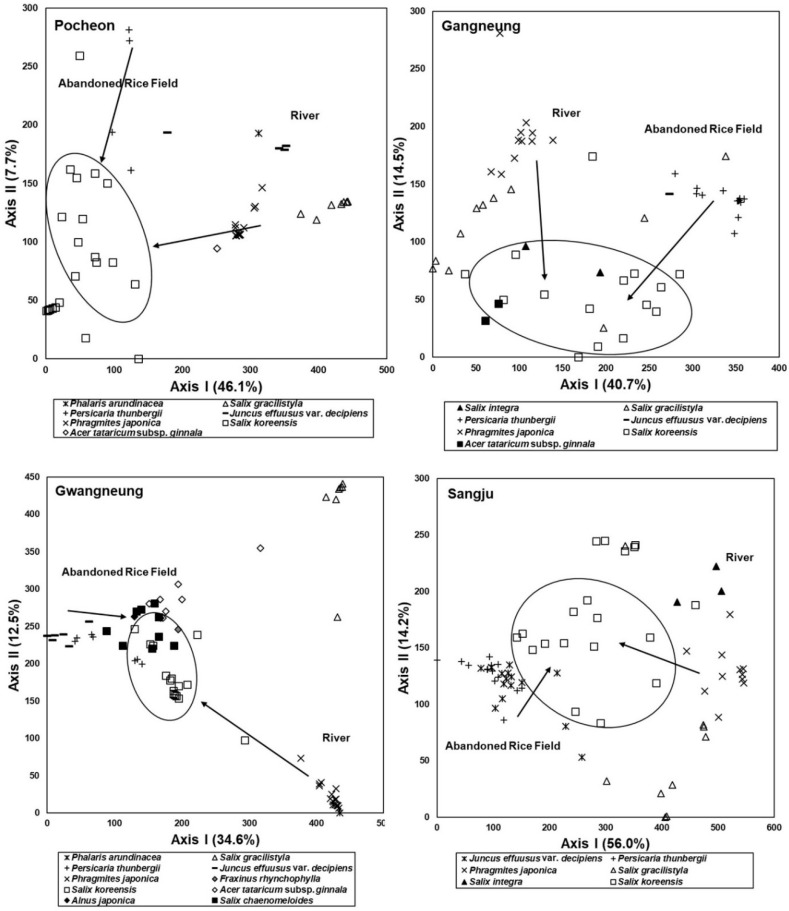
DCA ordination of stands based on vegetation data obtained from both abandoned rice fields and riparian zones of four rivers in watersheds where the rice fields are located. Arrangement of stands reflected successional change over years after abandonment and the sequence resembled the spatial distribution pattern of riparian vegetation appearing to recede from the waterfront in the riparian zone. ‘Abandoned Rice Field’ indicates stands surveyed in the rice fields and ‘River’ indicates stands surveyed in the riparian zones. The arrows show the successional trend in the abandoned rice fields and the changing trend of riparian vegetation according to the distance from the waterway. The stands in the part divided by solid lines represent the riparian forest stands established in the late successional stages of the abandoned rice fields and established at the farthest distance from the waterway in the riparian vegetation. They usually consist of a *S. koreensis* community, but they are mixed with *Salix chaenomeloids* community, *Acer tataricum* subsp. *ginnala* community, *Fraxinus rhynchophylla* community, and *A. japonica* community depending on the region.

**Figure 8 ijerph-19-10416-f008:**
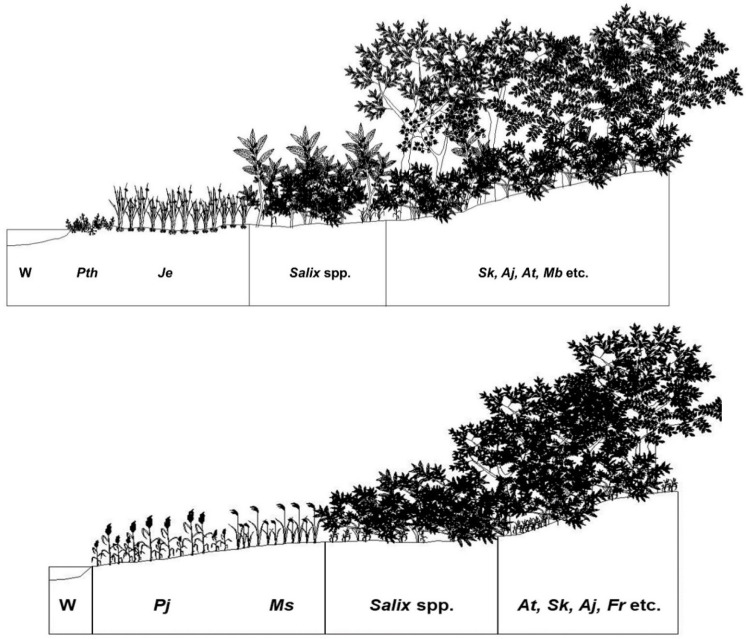
Reference information prepared by synthesizing the vegetation data collected from abandoned rice fields with different abandonment histories (upper) and riparian zones (lower). *Aj*: *Alnus japonica*; *At*: *Acer tataricum* subsp. *ginnala*; *Fr*: *Fraxinus rhynchophylla*; *Je*: *Juncus effusus* var. *decipiens*; *Mb*: *Morus bombycis*; *Ms*: *Miscanthus sacchariflorus*; *Pj*: *Phragmites japonica*; *Pth*: *Persicaria thunbergia*; *Salix* spp.: *Salix gracilistyla*, *Salix integra*, etc.; *Sk*: *Salix koreensis*; W: waterway.

**Table 1 ijerph-19-10416-t001:** The land use pattern in the watershed surrounding the midstream reach of the Hantan River, the upstream reach of the Geum River, the midstream reach of the Seomjin River, and the upstream reach of the Nakdong River.

River Name	Land Use Type (%)				
	Urbanized Area	Agricultural Field	Forest	Road	Bare Ground
Nakdong	0.9	14.3	80.4	4.4	0.0
Hantan	1.7	47.8	49.9	0.2	0.4
Geum	8.9	26.7	64.3	0.1	0.0
Seomjin	8.7	27.3	63.9	0.1	0.0

**Table 2 ijerph-19-10416-t002:** Geographical position of the reaches in which the riparian vegetation was investigated in four rivers selected for study.

River Name	Site No.	Site Name	Latitude	Longitude
Nakdong	1	Songjeong	37°03′54.10″	129°02′18.90″
	2	Gyeoldun bridge	37°00′56.00″	129°04′36.70″
	3	Seungbu station	36°59′31.20″	129°05′02.10″
	4	Wongok bridge	36°57′42.32″	129°05′27.71″
	5	Docheon	36°51′51.05″	128°54′13.76″
	6	Mt. Cheongnyang	36°46′45.62″	128°53′12.61″
	7	Andong Dam	36°44′05.01″	128°52′39.92″
Hantan	1	Yangji	38°15′02.3″	127°17′05.3″
Geum	1	Simcheon	36°12′17.4″	127°42′01.1″
Seomjin	1	Gurye	35°11′27.5″	127°32′47.3″

**Table 3 ijerph-19-10416-t003:** Dominant plant communities by successional stage in survey areas for abandoned rice paddies.

Survey Areas for Abandoned Rice Paddies	Dominant Plant Community by Successional Stage		
	Early Stage	Mid-Stage	Late Stage
Pocheon	*Persicaria thunbergii* community, *Juncus* *effuusus* var. *decipiens* community, and *Phalaris* *arundinacea* community	Young *Salix koreensis* community	Mature *S. koreensis* community
Gwangneung	*P. thunbergii* community and *J. effuusus* var. *decipiens* community	Not appeared	*S. koreensis* community, *S*. *chaenomeloides* community, *Alnus* *japonica* community, and *Acer tataricum* subsp. *ginnala* community
Asan	*P. thunbergii* community and *P. arundinacea* community	*Salix gracilistyla* community, *Salix integra* community, and young *S. koreensis* community	*S. koreensis* community and *S*. *chaenomeloides* community
Buyeo	*Persicaria longiseta* community, *Digitaria ciliaris* community, *Echinochloa utilis* community, *Bidens frondosa* community, *P. thunbergii* community, and *Leersia japonica* community	*S. integra* community, and young *S. koreensis* community	*S. koreensis* community
Naju	*P. thunbergii* community	*S. gracilistyla* community	*S. koreensis* community
Gangneung	*P. thunbergii* community and *J. effuusus* var. *decipiens* community	*S. gracilistyla* community and young *S. koreensis* community	*S. koreensis* community
Sangju	*P. thunbergii* community and *J. effuusus* var. *decipiens* community	*S. gracilistyla* community and young *S. koreensis* community	*S. koreensis* community
Pohang	*P. thunbergii* community and *Typha orientalis* community	*S. integra* community and young *S. koreensis* community	*S. koreensis* community

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
