# Peer review of "Succession of the Abandoned Rice Fields Restores the Riparian Forest"

_ijerph, 2022, doi:10.3390/ijerph191610416_

Round 1

Reviewer 1 Report

This article does a good job of describing the patterns of succession in abandon rice fields of South Korea based on a standard ordination technique. The research is important to potential restoration actions given the known linkages between riparian vegetation and other metrics of quality in river systems. The authors should include additional detail with respect to the statistical analyses, as choices used in conducting ordinations impact the results – this could simply be a statement of the packages used and versioning.  

 Line 4 – author list should have and before the final author’s name.

Propose revision

In order to study the relationship between the succession process of the abandoned rice paddies and riparian vegetation, information on riparian vegetation was collected in the same watersheds as the abandoned rice paddies being investigated.

We confirmed that the temporal sequence of vegetation change that occurred in the abandoned rice fields resembled the spatial distribution of the riparian vegetation.

Consequently, succession of the abandoned rice fields restored the riparian forest, which has almost disappeared in Korean and other Asian countries that depend on rice as their staple food.

Line 32 – delete  i.e. a comma suffices

Succession,  the sequential change in species composition or structure over time following a severe disturbance, has served as an organizing concept in ecology for more than a century [1,2].

Line 35  Old field succession, the sequence of change in plant communities on abandoned agricultural lands, is the subject of  a large body of ecological research [9- 36 12] that continues to yield new findings of interest to ecology community  (e.g., [8,13-22]).

Line 38 delete e.g. or use for example

Line 44 – proposed revision  – and a pattern of succession documented [29].

Line 92  -  proposed revision for clarity –

Finally, the aim of this study is to explore the feasibility of using information on vegetation succession from abandoned rice fields to identify reference conditions for restoring riparian vegetation.

Line 97  proposed revision

The abandoned rice paddies studied were selected from eight areas across the national territory of South Korea (Figure 1).

Line 98  proposed revision - … eastern regions, and then each divided into northern, central and southern regions and sites selected evenly from each of those six regions.

 Line 104.  The successional stages of the survey areas tended to be early stage, dominated by herbaceous plants, mid-stage, dominated by shrubby plants and young trees, and late stage, dominated by trees.

Line 106   Because abandonment of rice paddies began in areas that were inaccessible relative to residential areas, the successional states tended to be determined by distance from residential areas (Figure 2).

Line 117 – 139   This section seems more related to results and could likely be summarized more succinctly in a table.  Check the results for redundancy.

Line 140  Sites for surveying reference conditions for riparian vegetation were selected in …

Line 147  -  unclear  based on earlier descriptions it seems the order would be grassland, shrub-land and forest going towards the waterway, in other words the closer to the river the more forest. This sentence should be rewritten to make it clear.

…which appear in the order of grassland, shrub-land, and forest as far from the waterway.

Line 168 – 171

This information, number of plots by site could be included in a table, maybe combined with a table with the dominant species in each site at each stage of succession (currently described in lines 117-139) or could be included in Figure 1.

Line 180 – propose  …landscape were compared using ordination methods.

Line 181 – This sentence is a little awkward – profiles of vegetation strata were prepared using stand spread in 10 m belt transects from the abandoned rice field to the river.

Line 182  - transect should be plural transects.

Line 186  it is unclear what is meant by this sentence.

Therefore, in order to use the data as the reference information for the river restoration, it was systematized in contrast to the spatial distribution of the riparian vegetation

Line 195 – synthesizing both data – should be both datasets.

Line 197 unclear

Ordination was carried out to reveal the difference and similarity between species composition of plant communities occurring at different stages of succession (through time) and as a function of distance from the waterfront (through space).

Line 221.  tree forest -  could be simply .. maintain forest, which are composed of ….

Line 224  could be shortened - …   In the northwestern region stands tended to be arranged in the order of Juncus effuusus var. decipiens commu-nity, P. thunbergii community, Phalaris arundinacea community, and S. koreensis community as move from right to left parts on Axis l (Figure 4).

This same type of edit could be made in lines 235, 241, 251, 258, 268, 272  Currently each paragraph starts with some form of in the result…  that is redundant.  The results could be reported in a parallel form for each region, reducing the need to state they are results and that they are related to the abandoned rice fields – that is a given.  The next section provides a transition to talking about the reference sites and those comparisons.  This type of parallel reporting of results is done well in the next section - 2.3. Spatial distribution of the riparian vegetation based on stand profile.

Line 300 - There is inconsistency in using the italics for scientific names  - check the full article for that issue.

Line 324 – again each paragraph starts with some version of “as the result of” Lines 324, 330, 336. 340 could be edited to remove that redundancy.  Similarly with line 364, 383.

Line 378  should be     But stands established far away from the waterfront….

Line 378 – mention sites long abandoned being like those far from river (late successional) being close in ordination space.

 Line 402 – proposed revision -  Vegetation profiles from the grassland portions of the abandoned rice paddies and the riparian zone differed, but shrub dominated and tree dominated zones were similar.

This is not a major issue – but the word zone is used in reference to dominate vegetation (grassland, shrub, tree) and riparian.  Is there another word for one of those two cases?  Would sites be a better descriptor for the areas with grasslands, shrub or trees – since those vary where they occur on the landscape over time? 

Line 414  proposed revision -    A study on succession characterized successional patterns ……

A study on succession was begun from characterizing successional patterns describing the chronosequence of vegetation along sand dunes on the shores of Lake Michigan, moving from bare sand beach, to grasslands, to mature forests [61].

Line 430  This sentence is unclear.   Proposed revision - The study of succession, how biological community is reorganized after natural or artificial disturbances, has become the basis of ecology and theoretical frameworks that underpin many fields of the study [1,22,65,66].

Line 444 should be ….a variety of ways….

Line 445   That is, disturbance regimes vary in extent, spatial distribution, frequency, and recurrence interval depending on the environmental conditions [75=78].

Line 448   Rice fields are a kind of disclimax ….

Line 473   Historically, in Korea, abandonment of rice paddies was a rare phenomenon, but we were are able to find a number of rice paddies with different times since abandonment.

Line 489 – here the use of the word zone is applied to three parts of the riparian zone.  Maybe use of the word subzone when referring to the marginal, lower or upper?

Line 521 – introduce the use of zone 1 – it seems like this is referring to the marginal zone.  There are only three mentions of zone 1, zone 2  - it would be better to stick with marginal, lower, upper.

 In Korea, following the global trend, river restoration projects are being carried out, but the ecological quality of the streams has not improved as the principle of ecological restoration was not adhered to. In particular, the reference information, which should be a model in ecological restoration plans and a guideline for evaluation after restoration projects, is not utilized. Information on the ecological location of abandoned rice fields obtained from this study could contribute improved river restoration efforts in the future.  Furthermore, information on vegetation obtained in the process is expected to serve as reference information in the future, which will contribute to improving the quality of river restoration.

The study on succession, i.e., how biological community is reorganized after natural or artificial disturbances, has become the basis of ecology and theoretical frameworks underpin many fields of the study [1,22,65,66].

Author Response

Reply to the Reviewer 1

Thank you for your considerate review and kind comments for my manuscript. We sincerely answered your questions and comments, and the contents are as follows.

This article does a good job of describing the patterns of succession in abandon rice fields of South Korea based on a standard ordination technique. The research is important to potential restoration actions given the known linkages between riparian vegetation and other metrics of quality in river systems. The authors should include additional detail with respect to the statistical analyses, as choices used in conducting ordinations impact the results – this could simply be a statement of the packages used and versioning.  

 â˜ž We revised this part by reflecting reviewer’s comment. Lines 201-207

 Line 4 – author list should have and before the final author’s name.

☞ The authors were arranged by reflecting the contribution to the writing of the paper. Line 3.

Propose revision

In order to study the relationship between the succession process of the abandoned rice paddies and riparian vegetation, information on riparian vegetation was collected in the same watersheds as the abandoned rice paddies being investigated.

☞ We revised this part by reflecting reviewer’s comment. Lines 17-18.

We confirmed that the temporal sequence of vegetation change that occurred in the abandoned rice fields resembled the spatial distribution of the riparian vegetation.

☞ We’d like to maintain the current status to maintain connectivity with the front part.

Consequently, succession of the abandoned rice fields restored the riparian forest, which has almost disappeared in Korean and other Asian countries that depend on rice as their staple food.

 â˜ž We revised this part by referring reviewer’s comment. Line 27.

Line 32 – delete  i.e. a comma suffices

Succession,  the sequential change in species composition or structure over time following a severe disturbance, has served as an organizing concept in ecology for more than a century [1,2].

 â˜ž We revised this part by referring reviewer’s comment. Line 32.

Line 35  Old field succession, the sequence of change in plant communities on abandoned agricultural lands, is the subject of  a large body of ecological research [9- 36 12] that continues to yield new findings of interest to ecology community  (e.g., [8,13-22]).

  â˜ž We revised this part by referring reviewer’s comment. Line 36.

Line 38 delete e.g. or use for example

  â˜ž We revised this part by referring reviewer’s comment. Line 38

Line 44 – proposed revision  – and a pattern of succession documented [29].

  â˜ž We revised this part by referring reviewer’s comment. Line 44.

Line 92  -  proposed revision for clarity –

 Finally, the aim of this study is to explore the feasibility of using information on vegetation succession from abandoned rice fields to identify reference conditions for restoring riparian vegetation.

   â˜ž We revised this part by referring reviewer’s comment. Line 92.

Line 97  proposed revision

The abandoned rice paddies studied were selected from eight areas across the national territory of South Korea (Figure 1).

   â˜ž We revised this part by referring reviewer’s comment. Lines 98-99.

Line 98  proposed revision - … eastern regions, and then each divided into northern, central and southern regions and sites selected evenly from each of those six regions.

   â˜ž We revised this part by referring reviewer’s comment. Lines 98-102.

 Line 104.  The successional stages of the survey areas tended to be early stage, dominated by herbaceous plants, mid-stage, dominated by shrubby plants and young trees, and late stage, dominated by trees.

    â˜ž We revised this part by referring reviewer’s comment. Lines 105-108.

Line 106   Because abandonment of rice paddies began in areas that were inaccessible relative to residential areas, the successional states tended to be determined by distance from residential areas (Figure 2).

     â˜ž We revised this part by referring reviewer’s comment. Lines 108-110.

Line 117 – 139   This section seems more related to results and could likely be summarized more succinctly in a table.  Check the results for redundancy.

     â˜ž We summarized this part as Table 3 and moved to the Result section by referring reviewer’s comment. Table 3, Lines 235-236.

Line 140  Sites for surveying reference conditions for riparian vegetation were selected in …

     â˜ž We revised this part by referring reviewer’s comment. Line 142.

Line 147  -  unclear  based on earlier descriptions it seems the order would be grassland, shrub-land and forest going towards the waterway, in other words the closer to the river the more forest. This sentence should be rewritten to make it clear.

…which appear in the order of grassland, shrub-land, and forest as far from the waterway.

 â˜ž We revised this part by referring reviewer’s comment. Line 151.

Line 168 – 171

This information, number of plots by site could be included in a table, maybe combined with a table with the dominant species in each site at each stage of succession (currently described in lines 117-139) or could be included in Figure 1.

 â˜ž We’d like to maintain the current state according to the convention of related papers.

Line 180 – propose  …landscape were compared using ordination methods.

☞ Landscape was not been covered in this study and will be addressed in subsequent studies.

Line 181 – This sentence is a little awkward – profiles of vegetation strata were prepared using stand spread in 10 m belt transects from the abandoned rice field to the river.

 â˜ž We revised this part by referring reviewer’s comment. Lines 185-186.

Line 182  - transect should be plural transects.

☞ Yes, you are right. We also prepared several and chose the typical one among them. However, there were not many typical places in Korea, where land use in the riparian zone was very intense. In the case of abandoned rice fields, there was no significant variation among sites.

Line 186  it is unclear what is meant by this sentence.

Therefore, in order to use the data as the reference information for the river restoration, it was systematized in contrast to the spatial distribution of the riparian vegetation

☞ In abandoned rice paddies, the temporal sequence of vegetation that appears according to the successional stage showed a high similarity to the spatial distribution of riparian vegetation established by the frequency and intensity of flooding disturbance. Therefore, it means that the temporal sequence of the vegetation succession in abandoned rice paddies was systematized as reference information to restore vegetation when restoring rivers. We revised this part to contain this content. Lines 189-193.

Line 195 – synthesizing both data – should be both datasets.

 â˜ž We revised this part by referring reviewer’s comment. Line 202

Line 197 unclear

Ordination was carried out to reveal the difference and similarity between species composition of plant communities occurring at different stages of succession (through time) and as a function of distance from the waterfront (through space).

 â˜ž We revised this part by referring reviewer’s comment. Lines 205-206.

Line 221.  tree forest -  could be simply .. maintain forest, which are composed of ….

 â˜ž We revised this part by referring reviewer’s comment. Line 233.

Line 224  could be shortened - …   In the northwestern region stands tended to be arranged in the order of Juncus effuusus var. decipiens commu-nity, P. thunbergii community, Phalaris arundinacea community, and S. koreensis community as move from right to left parts on Axis l (Figure 4).

This same type of edit could be made in lines 235, 241, 251, 258, 268, 272  Currently each paragraph starts with some form of in the result…  that is redundant.  The results could be reported in a parallel form for each region, reducing the need to state they are results and that they are related to the abandoned rice fields – that is a given.  The next section provides a transition to talking about the reference sites and those comparisons.  This type of parallel reporting of results is done well in the next section - 2.3. Spatial distribution of the riparian vegetation based on stand profile.

 â˜ž We revised this part by referring reviewer’s comment. Lines 239-284.

Line 300 - There is inconsistency in using the italics for scientific names  - check the full article for that issue.

☞ We revised this part by referring reviewer’s comment.

Line 324 – again each paragraph starts with some version of “as the result of” Lines 324, 330, 336. 340 could be edited to remove that redundancy.  Similarly with line 364, 383.

 â˜ž We revised this part by referring reviewer’s comment. Lines 342-360.

Line 378  should be     But stands established far away from the waterfront….

  â˜ž We revised this part by referring reviewer’s comment. Lines 387-400.

Line 378 – mention sites long abandoned being like those far from river (late successional) being close in ordination space.

  â˜ž We revised this part to contain such a meaning by referring reviewer’s comment. Lines 387-400.

 Line 402 – proposed revision -  Vegetation profiles from the grassland portions of the abandoned rice paddies and the riparian zone differed, but shrub dominated and tree dominated zones were similar.

   â˜ž We revised this part by referring reviewer’s comment. Lines 433-436.

This is not a major issue – but the word zone is used in reference to dominate vegetation (grassland, shrub, tree) and riparian.  Is there another word for one of those two cases?  Would sites be a better descriptor for the areas with grasslands, shrub or trees – since those vary where they occur on the landscape over time? 

Different vegetation types of abandoned rice paddies represent successional stages, and different vegetation types of riparian vegetation represent toposequence established under the influence of different disturbance regime. So, we used ‘stage’ in the abandoned rice paddies and ‘zone’ in the riparian vegetation.

On the other hand, the disturbance regime is a long historical product, and it can be seen as the average of the long-term disturbances. The weather phenomenon fluctuates temporarily and the climate is the average it accumulated. Disturbance regime is the same as the climate. Therefore, it is not possible to change in a short time.

Line 414  proposed revision -    A study on succession characterized successional patterns ……

A study on succession was begun from characterizing successional patterns describing the chronosequence of vegetation along sand dunes on the shores of Lake Michigan, moving from bare sand beach, to grasslands, to mature forests [61].

☞The meaning is somewhat different, so I want to maintain the current state.

Line 430  This sentence is unclear.   Proposed revision - The study of succession, how biological community is reorganized after natural or artificial disturbances, has become the basis of ecology and theoretical frameworks that underpin many fields of the study [1,22,65,66].

   â˜ž We revised this part by referring reviewer’s comment. Lines 464-465.

Line 444 should be ….a variety of ways….

   â˜ž We revised this part by referring reviewer’s comment. Line 480.

Line 445   That is, disturbance regimes vary in extent, spatial distribution, frequency, and recurrence interval depending on the environmental conditions [75=78].

   â˜ž We revised this part by referring reviewer’s comment. Lines 481-482.

Line 448   Rice fields are a kind of disclimax ….

   â˜ž We revised this part by referring reviewer’s comment. Line 483.

Line 473   Historically, in Korea, abandonment of rice paddies was a rare phenomenon, but we were are able to find a number of rice paddies with different times since abandonment.

   â˜ž We revised this part by referring reviewer’s comment. Lines 509-511.

Line 489 – here the use of the word zone is applied to three parts of the riparian zone.  Maybe use of the word subzone when referring to the marginal, lower or upper?

   â˜ž We revised this part by referring reviewer’s comment. Lines 514-518.

Line 521 – introduce the use of zone 1 – it seems like this is referring to the marginal zone.  There are only three mentions of zone 1, zone 2  - it would be better to stick with marginal, lower, upper.

   â˜ž We revised this part by referring reviewer’s comment. Lines 555-568.

 In Korea, following the global trend, river restoration projects are being carried out, but the ecological quality of the streams has not improved as the principle of ecological restoration was not adhered to. In particular, the reference information, which should be a model in ecological restoration plans and a guideline for evaluation after restoration projects, is not utilized. Information on the ecological location of abandoned rice fields obtained from this study could contribute improved river restoration efforts in the future.  Furthermore, information on vegetation obtained in the process is expected to serve as reference information in the future, which will contribute to improving the quality of river restoration.

   â˜ž We revised this part by referring reviewer’s comment. 655-660.

Reviewer 2 Report

I found the paper very well contextualized and conceptualized. It certainly contributes to advancing the knowledge of plant ecology. Below are some of the suggestions for improvement.  

1.      I suggest authors present the location of the sites and different plant communities in different successional stages in a table (text from lines 109-151).

2.      It should be clearly mentioned how authors depicted early and mid-successional species composition (lines 188-189).

3.      Species names in Figure 3 caption and also in the text, for example, in lines 304-307 should be in italics. Please check this issue throughout the manuscript.

4.      The recommendations made in lines 392-396 should go in the later part of the manuscript somewhere in the discussion or conclusion.

5.      River restoration or riparian ecosystem restoration? Please clarify.   

Author Response

Reply to the Reviewer 2

Thank you for your considerate review and kind comments for my manuscript. We sincerely answered your questions and comments, and the contents are as follows.

I found the paper very well contextualized and conceptualized. It certainly contributes to advancing the knowledge of plant ecology. Below are some of the suggestions for improvement.  

  1. I suggest authors present the location of the sites and different plant communities in different successional stages in a table (text from lines 109-151).

 â˜ž We summarized this part as Table 3 by referring reviewer’s comment. Table 3, Lines 235-236.

  1. It should be clearly mentioned how authors depicted early and mid-successional species composition (lines 188-189)

 â˜ž We revised this part by referring reviewer’s comment. Line 190.

  1. Species names in Figure 3 caption and also in the text, for example, in lines 304-307 should be in italics. Please check this issue throughout the manuscript.

 â˜ž We revised this part by referring reviewer’s comment.

  1. The recommendations made in lines 392-396 should go in the later part of the manuscript somewhere in the discussion or conclusion.

 â˜ž We revised this part by referring reviewer’s comment. Lines 387-400.

  1. River restoration or riparian ecosystem restoration? Please clarify.   

☞ The river is a landscape, which is comprised of stream and riparian ecosystem. Between them, the stream ecosystem is a highly variable dynamic space. Therefore, it is a space that is difficult to apply restoration work, and it is highly likely that the effect will not be maintained even if applied. Therefore, in river restoration, actual restoration is performed in the riparian zone. Therefore, in fact, it is not an exaggeration to say that river restoration is a riparian restor
